# Comparison of 27-gauge and 25-gauge vitrectomy in the management of tractional retinal detachment secondary to proliferative diabetic retinopathy

Po-Lin Chen[1], Yan-Ting Chen[1,2,3], San-Ni Chen[1,4,5]*

1 Department of Ophthalmology, Changhua Christian Hospital, Changhua, Taiwan, 2 Institute of Clinical Medicine, National Yang-Ming University, Taipei, Taiwan, 3 Department of Optometry, Central Taiwan University of Science and Technology, Taichung, Taiwan, 4 College of Medicine, Chung-Shan Medical University, Taichung, Taiwan, 5 Department of Optometry, Da-Yeh University, Changhua, Taiwan

* 108562@cch.org.tw

**Data Availability Statement:** All relevant data are within the paper and its Supporting Information files.

## Abstract

### Objective

To compare surgical outcomes between 27 and 25-gauge vitrectomy in proliferative diabetic retinopathy (PDR) with tractional retinal detachment (TRD)

### Methods

This retrospective study was conducted to compare the intraoperative status, operation time, use of instruments, endotamponade substance, wound suture number, and iatrogenic break, between 27 and 25-gauge vitrectomy in 43 eyes afflicted by PDR with TRD. The post-surgical results, best-corrected visual acuity, intraocular pressure, recurrent vitreous haemorrhage, and re-operation rate were regularly followed up for 6 months.

### Results

Patients in the 25 and the 27-gauge groups did not differ significantly in terms of pre-surgical conditions, such as age, gender, pre-existing glaucoma, best-corrected visual acuity (BCVA) and the severity of their TRD. The mean operation time was 56.7 minutes in the 27-gauge group and 63.7 minutes in the 25-gauge group ($p = 0.94$). There is significantly less use of micro forceps in the 27-gauge group ($p = 0.004$). No difference between micro scissors and chandelier usage were noted; neither was their difference in iatrogenic retinal breaks. Significantly fewer wound sutures were noted in the 27-gauge group ($p < 0.001$). The post-operative results revealed no significant difference in ocular hypertension, hypotony, BCVA improvement, recurrent vitreous haemorrhage and re-operation rate.

### Conclusions

The 27-gauge vitrectomy system offers comparable surgical outcomes in PDR with TRD. The 27-gauge vitrectomy system is suitable for complicated retinal surgery.

**Funding:** The authors received no specific funding for this work.

**Competing interests:** The authors have declared that no competing interests exist.

## Introduction

The microincisional vitrectomy system (MIVS) was first introduced 30 years ago, and it gradually evolved from 23-gauge to 25-gauge and then, in recent years, to a 27-gauge system [1–4]. Though there may be disagreement between retinal surgeons regarding the advantages of a smaller gauge vitrectomy, there has been a continued trend toward using a smaller gauge vitrectomy system [4].

The development of the MIVS demonstrates several advantages over the traditional 20-gauge pars plana vitrectomy (PPV), such as self-sealing transconjunctival wounds, reduced corneal astigmatism, diminished sclerotomy tissue damage, reduced conjunctival scarring, decreased post-operative inflammation and reduced post-operative hypotony [5–9]. In addition, the powerful endoillumination, intraoperative intraocular pressure (IOP) stabilisation system, ultra-high-speed cutter and wide-angle viewing system have expanded the boundaries for using the MIVS [9].

Since the introduction of a 27-gauge vitrectomy system, there have been several studies comparing the 27-gauge and 25-gauge system in the management of retinal diseases, including rhegmatogenous retinal detachment (RRD), epiretinal membrane and proliferative diabetic retinopathy (PDR) [10–13]. In this retrospective study, we aim to compare the difference between the 27 and 25-gauge system in the management of tractional retinal detachment (TRD) secondary to PDR, to see whether the 27-gauge vitrectomy system offers advantages over the 25-gauge system in complicated retinal disease

## Materials and methods

In the retrospective, observational study, 48 consecutive eyes of TRD patients secondary to PDR who were undergoing vitrectomy using either 25- or 27-gauge vitrectomy system in the Changhua Christian Hospital between March 2018 and January 2019, http://www.cch.org.tw/about/about_7.aspx?id=0117214&sno=&dr_no=117214&pid=were reviewed. We initiated searching patients' medical records, identifying eligible patients, extracting the relevant data, and calculating the data from February 18th, 2019 to May 19th, 2020. One patient who did not complete a 6-month follow-up, two patients who had combined TRD and rhegmatogenous retinal detachment (CTRRD), and two patients who were uncooperative in undergoing complete ophthalmic examination, were excluded. Only 43 patients who were followed up postoperatively at the clinic for 6 months or more were included. All the surgeries were performed by one well-experienced retinal surgeon (SN Chen), and all the methods were carried out in accordance with relevant guidelines and regulations. This study was conducted under the approval number 191225 of the Ethical Committee of the Changhua Christian Hospital and the informed consents that clarified the permission of undergoing the surgery had been obtained from all of the patients. Other additional consents were waived by the ethics committee, because patient anonymity was maintained by the data source. All the medical records were collected since the surgery was done, between March 2018 and January 2019, and subsequent 6 months follow-up.

Patient data was recorded which included the duration of diabetes mellitus (DM), their HbA1c, age, gender, systemic disease, IOP, BCVA, the severity of TRD and pre-operative intravitreal injection (IVI) of anti-vascular endothelial growth factor (VEGF).

For the intraoperative status, we recorded the number of iatrogenic retinal breaks, iatrogenic cataract, simultaneous cataract surgery, the number of wound sutures, the substance of endotamponade (none, room air, SF6, C3F8, silicone oil), ancillary instrument usage for the removal of the fibrovascular membrane (such as micro forceps, micro scissors, chandelier lighting system; micro forceps used solely for macular pucker and internal limiting membrane

(ILM) removal was not included) and operation time. Operation time, recorded by the operation video, was defined as the time between insertion and removal of the trocar. In the cases with concomitant cataract surgery, we deducted the time to perform the cataract surgery from the total operation time.

Post-operative outcomes measures included the improvement of BCVA, ocular hypotony, ocular hypertension, the presence of recurrent vitreous haemorrhage (VH), re-operation rates, and time to develop cataract. Ocular hypotony is defined as an IOP of 6 mmHg or lower, and ocular hypertension is defined as an IOP of 25 mmHg or higher [12]. VH recurred at 1 month or longer after the surgery was recorded as recurrent haemorrhage [14]. Re-operation was confined to the eyes with recurrent retinal detachment or recurrent VH, excluding the cases of post-vitrectomy cataract surgery or silicone oil removal. Time to develop cataract was defined as longest to follow-up 6 months, excluding the cases of simultaneous cataract surgery or pre-operative cataract surgery.

## Staging of the severity of TRD

The severity of TRD was graded according to the previous literature as follows [15]:

- Grade I: Multiple-point adhesion with or without one site plaque-like broad adhesion

- Grade II: More than one broad adhesion, however, fewer than three sites located posterior to the equator

- Grade III: More than three broad adhesion sites, located posterior to the equator or extending beyond the equator within one quadrant

- Grade IV: Broad adhesions extending beyond the equator for more than one quadrant

## Vitrectomy and use of the surgical instrument

The majority of patients had IVI of Bevacizumab (Avastin®) (79.1%) and Ranibizumab (Lucentis®) (13.3%) within 7 days before operation to minimise intraoperative bleeding [16]. The patient received the vitrectomy surgery under retrobulbar and peribulbar anaesthesia. Vitrectomy was performed using the Alcon Constellation® Vision System (Alcon Laboratories, Fort Worth, Texas, USA). The cutting rate was 7500 cuts per min (cpm) in both the 25 and 27-gauge vitrectomy, and the aspiration pressure was set at 0–650 mm Hg. The sclerotomy of all eyes was incised at an angle of 30 degrees.

During the operation, core vitrectomy was performed first. The proliferative membrane was removed by using a vitrectomy probe, micro forceps or micro scissors. The vitrector was used as a vertical scissor for delamination when the space under the proliferative membrane allowed for the entry of the vitrector. Micro scissors and micro forceps were used when the membrane could not be delaminated and removed by the vitrector. Bimanual manoeuvres with the assistance of the chandelier lighting system were used for eyes with broad and tight adhesion, or when the retina became redundant secondary to iatrogenic breaks, that the membrane could not be efficiently removed with vitrector and scissors. For eyes with macular pucker, macular pucker and ILM were removed with micro forceps. Pan-retinal photocoagulation was applied by a curved laser probe. In eyes with iatrogenic retinal breaks, air-fluid exchange with or without the help of perfluorocarbon liquid was performed followed by gas or silicone oil tamponade. Scleral would be checked at the end of surgery for leakage. Sutures with 8–0 Vicryl was performed in eyes with wound leakage. Patients were given topical 1% prednisolone acetate and 0.5% levofloxacin four times a day and 0.5% atropine sulfate two times a day, postoperatively.

## Statistical analysis

Snellen BCVA was converted to logarithm of minimal angle of resolution (LogMAR) for calculation. Statistical analysis was performed using SPSS software (VER 15; SPSS, Chicago, IL). P values less than 0.05 were considered to be statistically significant.

Mann-Whitney U test was used for the comparison of numerical variables, including age, duration of DM, HbA1c, baseline BCVA, changes of BCVA, operation time, number of a wound suture and iatrogenic breaks between the two groups.

Fisher's exact test, Pearson chi-square test and Yate's continuity correction were used to compare the independent categorical variables in gender, pre-operative IVI of anti-VEGF, the severity of TRD, use of ancillary instruments, tamponade substances, ocular hypertension/hypotony, presence of recurrent VH and need for re-operation.

## Results

There were 43 eyes from 42 patients (22 males, 21 females) that were included in this study. There were 21 eyes in the 27-gauge groups and 22 eyes in the 25-gauge group. Preoperatively, the 27-gauge group and 25-gauge group did not differ significantly in age, gender, glaucoma history, grading of TRD severity, best-corrected visual acuity (BCVA) in LogMAR, HbA1c, duration of DM, pre-operative PRP, or IVI of ant-VEGF (bevacizumab/ranibizumab/none). Demographic data of patients is listed in **Table 1**.

As for the operative status, all the intraoperative data of patients is listed in **Table 2**. The 27-gauge and 25-gauge groups did not differ significantly in operation time (p = 0.94), though the operation time in the eyes of the 27-gauge group was shorter than the operation time in the 25-gauge group. The PRP spot number showed no significant difference between these two groups (p = 0.72). There is also no difference in the mean number of iatrogenic retinal breaks (p = 0.35), iatrogenic cataract (p = 0.49), simultaneous cataract surgery (p = 0.49), or the type of endotamponade substance between these two groups (p = 0.55). There was significantly lower rate of using micro forceps in the 27-gauge group (p = 0.004); however, there was no significant trend toward a lower rate of using micro scissors (p = 0.32) and chandelier lighting system (p = 0.52) in the 27-gauge group. At the end of the surgery, the eyes receiving 27-gauge vitrectomy rarely needed sutures for the sclerotomy wound (p = 0.0004).

Details of the post-operative status of the patients are shown in **Table 3**; there was no significant difference in the final BCVA and changes in BCVA between the 27-gauge and 25-gauge groups after vitrectomy, neither was their difference in post-operative ocular hypertension (p = 0.92) and ocular hypotony (p = 0.51). There was also no difference noted in recurrent VH (p = 1.0), incidence of re-operation related to the surgery (p = 0.26), and time to develop cataract (p = 0.86).

## Discussion

PDR is one of the major causes of visual impairment and eventual loss of sight in diabetic retinopathy patients. It can be further complicated by VH, fibrovascular membrane formation, TRD and CTRRD [17]. The treatment of TRD or CTRRD is challenging for the vitreoretinal surgeon. Generally, multiple strategies including segmentation, delamination and bimanual technique, are used to remove the fibrovascular membrane, and multiple ancillary instruments including micro forceps, micro scissors and bimanual techniques with chandelier lighting system are necessary for surgical success.

The development of the MIVS instrument demonstrates several aspects of advantages over the traditional PPV, like self-sealing transconjunctival wounds, reduced corneal astigmatism,

**Table 1. Demographic characteristics and clinical findings.**

| | 27 Gauge (%) (n = 21) | 25 Gauge (%) (n = 22) | Total (%) (n = 43) | *P* value |
|---|---|---|---|---|
| **Grade** | | | | |
| I | 2 (9.5) | 6 (27.3) | 8 (18.6) | 0.22 [a] |
| II | 6 (28.6) | 4 (18.2) | 10 (23.25) | |
| III | 6 (28.6) | 9 (40.9) | 15 (34.9) | |
| IV | 7 (33.3) | 3 (13.6) | 10 (23.25) | |
| **Age (year)** | | | | |
| Mean ± SD | 56.4 ± 8.5 | 50.3 ± 12.5 | 53.3 ± 11.2 | 0.21 [*] |
| Range | 32~68 | 29~65 | 29~68 | |
| **Underlying Disease** | | | | |
| Glaucoma | 1(4.8) | 3(13.6) | 4(9.3) | 0.61 [a] |
| **Gender** | | | | |
| Male | 10 (47.6) | 11 (50) | 21 (48.8) | 0.89 [b] |
| Female | 11 (52.4) | 11 (50) | 22 (51.2) | |
| **Pre-OP BCVA in LogMAR** | | | | |
| Mean ± SD | 1.43 ± 0.72 | 1.72 ± 0.69 | 1.58 ± 0.72 | 0.17 [*] |
| Range | 0.3~2.8 | 0.05~2.8 | 0.05~2.8 | |
| **Pre-OP IVI Anti-VEGF** | | | | |
| Bevacizumab (Avastin®) | 17 (80.95) | 17 (77.3) | 34 (79.05) | 1 [a] |
| Ranibizumab (Lucentis®) | 1 (4.75) | 2 (9.1) | 3 (7.0) | |
| None | 3 (14.3) | 3 (13.6) | 6 (13.95) | |
| **HbA1c** | | | | |
| Mean ± SD | 8.54 ± 1.97 | 8.68 ± 2.13 | 8.60 ± 2.04 | 0.88 [*] |
| Range | 5.7~14.2 | 6.1~12.7 | 5.7~14.2 | |
| **Duration of DM (year)** | | | | |
| Mean ± SD | 8.10 ± 5.85 | 9.80 ± 6.50 | 8.97 ± 6.18 | 0.95 [*] |
| Range | 1~20 | 0.5~20 | 0.5~20 | |
| **Pre-operative PRP [d]** | 6 (28.6) | 8 (36.4) | 14 (32.6) | 0.82 [b] |

The table mention the pre-surgical condition of 27-gauge, 25-gauge, and their comparison.

**Abbreviations**: SD: standard deviation, Pre-OP: pre-operative, BCVA: best corrected visual acuity, LogMAR: logarithm of minimal angle of resolution, IVI: intravitreal injection, VEGF: vascular endothelial growth factor, HbA1c: hemoglobin A1c, DM: diabetes mellitus, PRP: pan-retinal photocoagulation.

[*]: Mann-Whitney U test

a: Fisher's exact test

b: Pearson chi-square test.

d: Pre-operative PRP and intra-operative PRP spot number showed no statistical significance; thus, they didn't make significant influence in the operation time.

less sclerotomy-related tissue damage, less conjunctival scarring, decreased post-operative inflammation, reduced post-operative hypotony and endophthalmitis [5–8].

Owing to the improved and powerful endoillumination, intraoperative IOP stabilisation system, ultra-high-speed cutter and wide-angle viewing system, the utilisation of a 27-gauge vitrectomy system is possible in complicated vitrectomy cases [9]. In this comparative study, we also showed a comparable surgical outcome achieved in the 25 and 27-gauge groups.

In this study, a significantly fewer scleral wound suture was noted. Previous literatures had shown the similar trends [13, 18–20], and our study further demonstrated a statistical significance. Since in the situation of PDR with TRD, the longer surgical time for fibrotic tissue removal and thus stretching more on scleral wound, which affects wound integrity more, a smaller 27-gauge wound can better demonstrate the non-suture technique in cases of PDR

**Table 2. Operative status.**

|  | 27 Gauge (%) (n = 21) | 25 Gauge (%) (n = 22) | Total (%) (n = 43) | *P* value |
|---|---|---|---|---|
| **OP time (min)** |  |  |  |  |
| Mean ± SD | 56.7 ± 19.6 | 63.7 ± 39.4 | 60.6 ± 30.9 | 0.94 * |
| Range | 33~106 | 13~170 | 13~170 |  |
| **Use of instruments** |  |  |  |  |
| Micro forceps | 14 (66.7) | 22 (100) | 36 (83.7) | **0.004** [a] |
| Micro scissors | 1 (4.8) | 3 (13.6) | 4 (9.3) | 0.32 [a] |
| Chandelier | 2 (9.5) | 3 (13.6) | 5 (11.6) | 0.52 [a] |
| **Endotamponade substance** |  |  |  |  |
| None | 7 (33.3) | 7 (31.8) | 14 (32.6) | 0.55 [a] |
| Room air | 0 (0) | 2 (9.1) | 2 (4.7) |  |
| SF6 | 6 (28.6) | 5 (22.7) | 11 (25.6) |  |
| C3F8 | 5 (23.8) | 3 (13.6) | 8 (18.6) |  |
| Silicone oil | 3 (14.3) | 5 (22.7) | 8 (18.6) |  |
| **Wound suture number** |  |  |  |  |
| Mean ± SD | 0.05 ± 0.21 | 1.27 ± 1.39 | 0.67 ± 1.18 | **0.0004** * |
| Range | 0~1 | 0~4 | 0~4 |  |
| **Iatrogenic break** |  |  |  |  |
| Mean ± SD | 1.05 ± 2.13 | 0.86 ± 1.32 | 0.95 ± 1.77 | 0.35 * |
| Range | 0~9 | 0~4 | 0~9 |  |
| **Iatrogenic cataract** [e] | 1 (4.8) | 0 (0) | 1 (2.3) | 0.49 [a] |
| **Simultaneous cataract OP** [e] | 1 (4.8) | 0 (0) | 1 (2.3) | 0.49 [a] |
| **PRP spots number** [d] |  |  |  |  |
| Mean ± SD | 1887 ±1347 | 1836 ± 989 | 1861 ± 1192 | 0.72 * |
| Range | 482~6654 | 0~4351 | 0~6654 |  |

The table mention the intra-operative outcomes of 27-gauge, 25-gauge, and their comparison.

**Abbreviations**: OP: Operation, min: minute, SD: Standard Deviation, SF6: Sulfur hexafluoride, C3F8: Octafluoropropane, PRP: pan-retinal photocoagulation.

*: Mann-Whitney U test

a: Fisher's exact test.

d: Pre-operative PRP and intra-operative PRP spot number showed no statistical significance; thus, they didn't make significant influence in the operation time.

e: One patient in severity grade 4 of TRD received simultaneous cataract surgery, and the other one received cataract surgery due to iatrogenic cataract. Both cases have no much influence on the result of visual acuity improvement.

with TRD. Less need of scleral sutures should be especially beneficial for wound healing in the eyes with PDR. Since those eyes are inclined to suffer from corneal epithelial defect either intraoperatively or postoperatively, a smooth ocular surface without suture irritation can help faster recovery of ocular surface. In addition to the universal benefits of smaller wound size, less wound leakage and less post-operative discomfort, we also noticed less use of ancillary instruments in the 27-gauge group. Patients in the 27-gauge group significantly reduce the need for using micro forceps; they can also accomplish the operation without the demands of micro scissors and chandelier lighting system, though without statistical significance. Because of the improved fluidics of 27-gauge system, smaller sphere of influence, the vitrector could partially be served as a micro forceps by engaging the margin of the fibrovascular membrane with suction, and we can perform membrane peeling without micro forceps under some circumstances. Besides, because of the small size of 27-gauge vitrector, there is more chance of the vitrector to gain access into the space between the tissue planes and can be served as a vertical scissors in tissue delamination without exerting much tractional force and making

**Table 3. Post-surgical results.**

| | 27 Gauge (%) (n = 21) | 25 Gauge (%) (n = 22) | Total (%) (n = 43) | *P* value |
|---|---|---|---|---|
| **Post-OP** BCVA in LogMAR | | | | |
| **Mean ± SD** | 0.79 ± 0.55 | 0.94 ± 0.69 | 0.87 ± 0.63 | 0.58 * |
| **Range** | 0~1.7 | 0~2.8 | 0~2.8 | |
| **BCVA improving in LogMAR** | | | | |
| **Mean ± SD** | 0.60 ± 0.67 | 0.80 ± 0.61 | 0.70 ± 0.65 | 0.22 * |
| **Range** | -0.4~2.08 | -0.02~2.3 | -0.4~2.3 | |
| **IOP** | | | | |
| **Ocular hypertension** | 8 (38.1) | 7 (31.8) | 15 (34.9) | 0.92 c |
| **Ocular hypotony** | 0 (0) | 1 (4.5) | 1 (2.3) | 0.51 a |
| **Total** | 8 (38.1) | 8 (36.4) | 16 (37.2) | 0.84 c |
| **Recurrent VH** | 0 (0) | 0 (0) | 0 (0) | 1.00 a |
| **Re-operation** | 0 (0) | 2 (9.1) | 2 (4.7) | 0.26 a |
| **Time to develop cataract** f | | | | |
| **Cataract patient/total patient** | 3/13 | 3/18 | 6/31 | 0.50 a |
| **Mean ± SD (month)** | 5.62 ± 0.77 | 5.28 ± 1.71 | 5.42 ± 1.39 | 0.86 * |
| **Range (month)** | 4~6 | 1~6 | 1~6 | |

The table mention the post-surgical outcomes of 27-gauge, 25-gauge, and their comparison.

**Abbreviations**: Post-OP: post-operative, BCVA: best corrected visual acuity, LogMAR: logarithm of minimal angle of resolution, SD: standard deviation, IOP: intraocular pressure, VH: vitreous hemorrhage.

*: Mann-Whitney U test

a: Fisher's exact test

c: Yate's continuity correction of Pearson chi-square test.

f: In 27-gauge group, there are six cases received preoperative cataract surgery, and two cases received intraoperative cataract surgery; in 25-gauge group, there are four cases received preoperative cataract surgery.

iatrogenic breaks on the adjacent retina. In addition, it can be used for blunt dissection to loosen the adhesive tissues and be served as a pick to elevate tissue. With the multi-functionality of the vitrector, there would be fewer exchanges of instruments, which can shorten the surgical time and the likelihood of iatrogenic breaks. Theoretically, this minimizes the need for micro scissors and the bimanual dissection technique; however, we didn't show the statistical significance of using micro scissors and chandelier light. This can be explained by the limited case number and that we didn't further calculate the total number of times we exchanged the instruments within each surgery. However, the vitrector still cannot totally replace the role of micro forceps in the peeling of epiretinal membrane and internal limiting membrane. Besides, in the situation of extremely adherent fibrotic membrane, micro scissors and chandelier lighting system are still necessary in 27-gauge system.

In this comparative study, we noted that the operation time is similar in both groups, if not shorter in the 27-gauge group, which is different from previous reports in which more time is needed in the 27-gauge vitrectomy system in the management epiretinal membrane and PDR [10–12]. The difference from previous reports may lie in the fact that eyes with PDR and TRD generally have more liquefied vitreous, thus, the difference of the surgical time for removing the core vitreous between these two systems is less. Moreover, the use of the 27-gauge system allows for lower rates of instrument exchange and therefore, the removal of the fibrotic membrane is more efficient. This shortens the time for membrane removal. A recently published study [12, 20, 21] also showed that there was no difference in operation time for vitrectomy between 27-gauge and 25-gauge vitrectomy systems in the management of RRD. Eyes with

RRD, as eyes in PDR with TRD, have a more liquefied vitreous. Core vitrectomy, thus, is no longer the most time-consuming procedure. As a result of the smaller sphere of influence in the 27-gauge vitrectomy system, the shaving of the vitreous gel near the detached retina is much safer and efficient. Henceforth, it is not surprising that the surgical time is not longer in the eyes of PDR with TRD managed with the 27-gauge system.

There are some limitations of this study. First, this is a retrospective study. Second, the case number is small. There is also some power in this study, that all the surgeries were performed by one surgeon, which alleviates the difference of surgical techniques from different surgeons.

## Conclusion

In conclusion, the 27-gauge vitrectomy offered comparable surgical results, with less ancillary instrument usage, fewer wound sutures, and similar surgical time. Our study showed the non-inferiority of the 27-gauge over the 25-gauge vitrectomy system in complicated retinal diseases of TRD in PDR patients. However, further prospective study with a larger case number is necessary to validate our conclusion.

## Supporting information

**S1 Data.**
(XLSX)

## Acknowledgments

I thank the editors from Enago for their expertise and assistance throughout all aspects of our study and for their help in writing the manuscript.

## Author Contributions

**Conceptualization:** Po-Lin Chen, San-Ni Chen.

**Data curation:** Po-Lin Chen, Yan-Ting Chen, San-Ni Chen.

**Formal analysis:** Po-Lin Chen, Yan-Ting Chen, San-Ni Chen.

**Investigation:** Po-Lin Chen, Yan-Ting Chen, San-Ni Chen.

**Methodology:** Po-Lin Chen, Yan-Ting Chen, San-Ni Chen.

**Project administration:** San-Ni Chen.

**Resources:** San-Ni Chen.

**Supervision:** Yan-Ting Chen, San-Ni Chen.

**Validation:** Po-Lin Chen, San-Ni Chen.

**Visualization:** Po-Lin Chen, San-Ni Chen.

**Writing – original draft:** Po-Lin Chen, San-Ni Chen.

**Writing – review & editing:** Po-Lin Chen, San-Ni Chen.

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
