## [Decision Letter · Decision Letter 0]

7 Dec 2020

PONE-D-20-33300

Comparison of 27-gauge and 25-gauge vitrectomy in the management of tractional retinal detachment secondary to proliferative diabetic retinopathy

PLOS ONE

Dear Dr. Chen,

Thank you for submitting your manuscript to PLOS ONE. After careful consideration, we feel that it has merit but does not fully meet PLOS ONE’s publication criteria as it currently stands. Therefore, we invite you to submit a revised version of the manuscript that addresses the points raised during the review process.

We look forward to receiving your revised manuscript.

Kind regards,

Michael Mimouni

Academic Editor

PLOS ONE

Journal Requirements:

2.We note that you have indicated that data from this study are available upon request. PLOS only allows data to be available upon request if there are legal or ethical restrictions on sharing data publicly. For information on unacceptable data access restrictions, please see http://journals.plos.org/plosone/s/data-availability#loc-unacceptable-data-access-restrictions.

4.Please provide additional details regarding participant consent. In the ethics statement in the Methods and online submission information, please ensure that you have specified what type you obtained (for instance, written or verbal, and if verbal, how it was documented and witnessed). If your study included minors, state whether you obtained consent from parents or guardians.

In addition, Please clarify the context in which consent was obtained, and specify whether patients provided:

    1) Consent to use their medical records/samples used in research

    2) Consent to undergo the procedure

    3) Consent to take part in the study reported in this manuscript.

If the need for additional consent was waived by the ethics committee, please include this information.

5. In the ethics statement in the manuscript and in the online submission form, please provide additional information about the patient records/samples used in your retrospective study, including: a) whether all data were fully anonymized before you accessed them; b) the date range (month and year) during which patients' medical records/samples were accessed.

Reviewers' comments:

Reviewer's Responses to Questions

**Comments to the Author**

1. Is the manuscript technically sound, and do the data support the conclusions?

Reviewer #1: Yes

Reviewer #2: Yes

2. Has the statistical analysis been performed appropriately and rigorously? 

Reviewer #1: Yes

Reviewer #2: I Don't Know

3. Have the authors made all data underlying the findings in their manuscript fully available?

Reviewer #1: No

Reviewer #2: Yes

4. Is the manuscript presented in an intelligible fashion and written in standard English?

Reviewer #1: No

Reviewer #2: No

5. Review Comments to the Author

Reviewer #1: Review:

PONE-D-20-33300

Comparison of 27-gauge and 25-gauge vitrectomy in the management of tractional retinal detachment secondary to proliferative diabetic retinopathy

This study was well conducted, and the manuscript is well written, but there are several points of concern. First, a retrospective small sample size of 43 patients. Secondly, the findings of this study are not novel and there more than a few previous studies to compare 27 gauge to 25-gauge vitrectomy for RRD and TRD.

Few minor points:

1. Randomization: How patients were randomized for both groups? If there were no randomization how the author explains the similarity between both groups?

2. Lines 135-137: “There was significantly lower rate of using micro forceps in the 27-gauge group (p = 0.004), and a trend toward a lower rate of using micro scissors (p = 0.32) and chandelier lighting system (p = 0.52) in the 27-gauge group.” P=0.32 and 0.52 is not a trend. Indeed, the usage of 27 gauge instead of 25 gauge in theory should make surgeon use less intra ocular devices, but the author did not demonstrate it in this current study.

Reviewer #2: Comparison of 27-gauge and 25-gauge vitrectomy in the management of tractional retinal detachment secondary to proliferative diabetic retinopathyMaterials and Methods

Here are my comments:

1) In the methods and materials, you’ve discussed the status of pre-operative treatment including Anti-VEGF, you have not mentioned the pre-operative treatment with laser pan photocoagulation PRP which usually shorten the time of surgery – if previously done- and also make the vitrectomy easier for some extent

2) Regarding post-operative BCVA improvement, how did you define the improvement especially in cases with combined PPV and cataract surgery?

3) Have you noticed any significant difference in post vitrectomy cataract development between 25G and 27G?

4) As the cutter and the light pipe are thinner in 27G, have you noticed more bending of the instruments in 27G than that of 27G, or there was no significant difference?

6. PLOS authors have the option to publish the peer review history of their article (what does this mean?). If published, this will include your full peer review and any attached files.

Reviewer #1: No

Reviewer #2: No

---

## [Author Response · Author response to Decision Letter 0]

25 Jan 2021

Response to Reviewers

We thank all the reviewers for giving us the suggestions and opportunity to submit a revised manuscript and hope that this improved manuscript is acceptable for publication in the journal of PLOS ONE®.

Response to Reviewer # 1:

1. Comment: A retrospective small sample size of 43 patients

Author response: Thank you for pointing this out. Because of the improved care of diabetes mellitus in Taiwan, the number of TRD secondary to PDR is decreasing, besides, we only enroll cases of TRD with macular involvement, which explained the limited case number. However, it is beyond denial; this is a weak point of this study, and we had mentioned the paragraph discussing about the limitations of the study.

2. Comment: The findings of this study are not novel and there more than a few previous studies to compare 27 gauge to 25-gauge vitrectomy for RRD and TRD.

Author response: Most of the previous comparative studies of 27- and 25-gauge vitrectomy were discussing about RRD. I believe there are still some findings in this manuscript that can support us to manage more complicated retinal disease like TRD secondary to PDR by choosing 27-gauge vitrectomy.

3. Comment: How patients were randomized for both groups? If there were no randomization how the author explains the similarity between both groups?

Author response: Since this is a retrospective study, there was no randomization. The similarity between these two groups were because all the cases of 27-gauge were cases having the surgery after August 2018 (the cases were collected from March 2018 to January 2019), when 27-gauge vitrectomy was available in our hospital, before that, we use 25-gauge vitrectomy for PDR with TRD. Since those patients were all DM with TRD. It is of no surprise patients of this disease had similar background conditions and similar demographic data. 

4. Comment: Lines 135-137: “There was significantly lower rate of using micro forceps in the 27-gauge group (p = 0.004), and a trend toward a lower rate of using micro scissors (p = 0.32) and chandelier lighting system (p = 0.52) in the 27-gauge group.” p = 0.32 and 0.52 is not a trend. Indeed, the usage of 27 gauge instead of 25 gauge in theory should make surgeon use less intra ocular devices, but the author did not demonstrate it in this current study.

Author response: Though 27G vitrector can significantly reduce the use of micro forceps. We still cannot show the trend of less use in micro scissors and chandelier lighting system, though we did feel that the timing of using micro scissors is less. The possible explanation is that, first, the case number is not large enough, which could not show statistical significance, and second, though we still use micro scissors in 27-gauge system in some complicated cases, the total times in each surgery using micro scissor may still be less, however, we need another study to confirm this. In the revised manuscript, I will change the sentence of “trend” to “no significant trend” (Lines 140-141), instead, we will discuss why we did not show significantly less usage of micro scissors in this study (Lines 184-188). 

Response to Reviewer # 2:

1. Comment: In the methods and materials, you’ve discussed the status of pre-operative treatment including Anti-VEGF, you have not mentioned the pre-operative treatment with laser pan photocoagulation PRP which usually shorten the time of surgery – if previously done- and also make the vitrectomy easier for some extent

Author response: Thank you for the reviewer point out the weak point of this study. In the revised manuscript, the information of previous PRP will be added. We will add the pre-operative PRP, intra-operative PRP spot number (Lines 136-137, Table 1 and 2), and thus, this PRP number cannot influence total operation time (Lines 299-300, Line 308-309).

2. Comment: Regarding post-operative BCVA improvement, how did you define the improvement especially in cases with combined PPV and cataract surgery?

Author response: There are only two cases had received simultaneous cataract surgery, without statistical significance. One of them is in the severity stage 4 of TRD, and cataract is not an important factor to influence the visual acuity improvement. The other one received cataract surgery due to intra-operative iatrogenic cataract, and this cannot influence the visual acuity improvement. We added the simultaneous cataract surgery (Lines 138-139, Table 2) and further discussed this. (Lines 306-308)

3. Comment: Have you noticed any significant difference in post vitrectomy cataract development between 25G and 27G ?

Author response: In the 27-gauge group, there were 10 cases receiving cataract surgery 6 months later or longer after, and 3 cases received cataract surgery at 4, 4, 5 months. While in the 25-gauge group, there were 15 cases receiving cataract surgery 6 months later or longer after, and 3 cases received cataract surgery at 1, 1, 3 months. There are no significant difference between 2 groups in cataract development. We will add time to develop cataract. (Lines 315-317, Table3)

4. Comment: As the cutter and the light pipe are thinner in 27G, have you noticed more bending of the instruments in 27G than that of 27G, or there was no significant difference?

Author response: We did notice there was more chance of instrument bending in the 27-gauge system when we initially use 27-gauge system. However, after a short learning curve, bending of instrument seldom happens. To avoid bending of instruments, you should not rotate the eye ball too much with the instrument as we used to do with 25- or 23-gauge system. To address the far peripheral lesions, use indentation, instead of rotating the eye ball.

---

## [Decision Letter · Decision Letter 1]

12 Mar 2021

Comparison of 27-gauge and 25-gauge vitrectomy in the management of tractional retinal detachment secondary to proliferative diabetic retinopathy

PONE-D-20-33300R1

Dear Dr. Chen,

We’re pleased to inform you that your manuscript has been judged scientifically suitable for publication and will be formally accepted for publication once it meets all outstanding technical requirements.

Kind regards,

Michael Mimouni

Academic Editor

PLOS ONE

Additional Editor Comments (optional):

PLOS ONE does not reject manuscripts based on perceived impact or significance. Therefore, the manuscript is suitable for publication. 

Reviewers' comments:

Reviewer's Responses to Questions

**Comments to the Author**

1. If the authors have adequately addressed your comments raised in a previous round of review and you feel that this manuscript is now acceptable for publication, you may indicate that here to bypass the “Comments to the Author” section, enter your conflict of interest statement in the “Confidential to Editor” section, and submit your "Accept" recommendation.

Reviewer #1: All comments have been addressed

Reviewer #2: All comments have been addressed

2. Is the manuscript technically sound, and do the data support the conclusions?

Reviewer #1: Yes

Reviewer #2: Yes

3. Has the statistical analysis been performed appropriately and rigorously? 

Reviewer #1: Yes

Reviewer #2: Yes

4. Have the authors made all data underlying the findings in their manuscript fully available?

Reviewer #1: No

Reviewer #2: Yes

5. Is the manuscript presented in an intelligible fashion and written in standard English?

Reviewer #1: Yes

Reviewer #2: Yes

7. PLOS authors have the option to publish the peer review history of their article (what does this mean?). If published, this will include your full peer review and any attached files.

Reviewer #1: No

Reviewer #2: No

6. Review Comments to the Author

Reviewer #1: This study is well designed, and the manuscript is well written. But, unfortunately, this study has a very small sample size, without novel or significant findings.

Reviewer #2: (No Response)

---

## [Editor Report · Acceptance letter]

16 Mar 2021

PONE-D-20-33300R1 

Comparison of 27-gauge and 25-gauge vitrectomy in the management of tractional retinal detachment secondary to proliferative diabetic retinopathy 

Dear Dr. Chen:

I'm pleased to inform you that your manuscript has been deemed suitable for publication in PLOS ONE. Congratulations! Your manuscript is now with our production department. 

Kind regards, 

on behalf of

Dr. Michael Mimouni 

Academic Editor

PLOS ONE